# The chemokine receptor CX$_3$CR1 coordinates monocyte recruitment and endothelial regeneration after arterial injury

Tobias Getzin[1,‡,†], Kashyap Krishnasamy[1,2,†], Jaba Gamrekelashvili[1,2], Tamar Kapanadze[1,2], Anne Limbourg[1,§], Christine Häger[1,¶], L Christian Napp[1,3], Johann Bauersachs[3], Hermann Haller[2] & Florian P Limbourg[1,2,*] [iD]

## Abstract

Regeneration of arterial endothelium after injury is critical for the maintenance of normal blood flow, cell trafficking, and vascular function. Using mouse models of carotid injury, we show that the transition from a static to a dynamic phase of endothelial regeneration is marked by a strong increase in endothelial proliferation, which is accompanied by induction of the chemokine CX$_3$CL1 in endothelial cells near the wound edge, leading to progressive recruitment of Ly6C$^{lo}$ monocytes expressing high levels of the cognate CX$_3$CR1 chemokine receptor. In *Cx3cr1*-deficient mice recruitment of Ly6C$^{lo}$ monocytes, endothelial proliferation and regeneration of the endothelial monolayer after carotid injury are impaired, which is rescued by acute transfer of normal Ly6C$^{lo}$ monocytes. Furthermore, human non-classical monocytes induce proliferation of endothelial cells in co-culture experiments in a VEGFA-dependent manner, and monocyte transfer following carotid injury promotes endothelial wound closure in a hybrid mouse model *in vivo*. Thus, CX$_3$CR1 coordinates recruitment of specific monocyte subsets to sites of endothelial regeneration, which promote endothelial proliferation and arterial regeneration.

**Keywords** CX$_3$CR1; endothelial cells; monocytes; regeneration; vascular injury
**Subject Categories** Cardiovascular System; Immunology

## Introduction

Maintenance and restoration of endothelial integrity are critical for blood vessel function. Endothelial cells (EC) form a monolayer in the inner surfaces of blood vessels that controls exchange of metabolites and regulates coagulation and cell trafficking. Cardiovascular diseases, such as atherosclerosis, vascular interventions, or bypass surgery, cause EC damage or overt defects in the endothelial monolayer, which triggers vascular inflammation, neointima formation, and ultimately vessel obstruction if endothelial integrity is not restored (Gimbrone & Garcia-Cardena, 2016).

Under physiological conditions, EC replication is inhibited by cell contact and laminar flow (Akimoto *et al*, 2000; Chen *et al*, 2000). The loss of few cells is repaired rapidly by extension and spreading of adjacent EC without the need for proliferation (Reidy & Schwartz, 1981). However, larger EC lesions require proliferation to regenerate the endothelial monolayer and prevent neointima formation (Haudenschild & Schwartz, 1979). *In vivo*, proliferation is initiated near the wound edge but subsequently spreads to areas distant from the wound (Filipe *et al*, 2008), which contrasts with *in vitro* EC wound models, in which the zone of proliferation is narrow and restricted to the immediate wound edge (Chen *et al*, 2000), suggesting the contribution of so far unrecognized cellular interactions and mechanisms in the coordination of EC proliferation *in vivo*.

Vascular injury leads to a regulated inflammatory response, during which neutrophils and monocytes are recruited by chemokines (Koenen & Weber, 2011). The CX$_3$CL1-CX$_3$CR1 axis is a critical regulator of the vascular injury response (Schafer *et al*, 2004; Flierl *et al*, 2015). CX$_3$CL1 (fractalkine) is a membrane-bound chemokine expressed by activated EC after injury (Bazan *et al*, 1997; Liu *et al*, 2006), which binds to its cognate receptor CX$_3$CR1 expressed on

1 Vascular Medicine Research, Hannover Medical School, Hannover, Germany
2 Department of Nephrology and Hypertension, Hannover Medical School, Hannover, Germany
3 Department of Cardiology and Angiology, Hannover Medical School, Hannover, Germany
*Corresponding author. Tel: +49 5115 329589; E-mail: limbourg.florian@mh-hannover.de
†These authors contributed equally to this work
‡Present address: Institute of Radiology, Hannover Medical School, Hannover, Germany
§Present address: Department of Plastic, Aesthetic, Hand and Reconstructive Surgery, Hannover Medical School, Hannover, Germany
¶Present address: Institute for Laboratory Animal Science and Central Animal Facility, Hannover Medical School, Hannover, Germany

monocytes, subsets of NK cells and T cells (Jung et al, 2000), and which mediates preferential arrest of monocytes and adhesion to EC under flow (Fong et al, 1998; Ancuta et al, 2003).

A minor subset of monocytes with patrolling behavior, called Ly6C$^{lo}$ monocytes in mice and which corresponds to the non-classical subset of human monocytes, shows high levels of CX$_3$CR1 and interacts constitutively with blood vessels, where under high flow conditions, this subset is preferentially recruited to activated EC by CX$_3$CL1 (Ancuta et al, 2003; Schulz et al, 2007). In the steady state, Ly6C$^{lo}$ monocytes originate from inflammatory Ly6C$^{hi}$ monocytes in a process called "monocyte conversion" (Yona et al, 2013; Gamreke-lashvili et al, 2016). They are long-lived and crawl along the luminal side of EC to monitor blood vessels and scavenge microparticles (Auffray et al, 2007), a feature shared with the human non-classical monocyte subset (Cros et al, 2010). In kidney models of endothelial injury, Ly6C$^{lo}$ monocytes orchestrate EC clearance through interaction with subsets of activated EC in a CX$_3$CL1- and CX$_3$CR1-dependent manner (Carlin et al, 2013). Ly6C$^{lo}$ monocytes were also suggested to contribute to ischemic tissue repair (Nahrendorf et al, 2007). Of note, monocytes have recently been reported to induce EC proliferation in vitro (Schubert et al, 2008) and to participate in vascular repair after vascular injury in vivo (Becher et al, 2014). Furthermore, Ly6C$^{lo}$ monocytes confer endothelial protection in atherosclerosis models (Quintar et al, 2017).

We studied the relevance of CX$_3$CR1 for monocyte subset recruitment and endothelial regeneration in a model of perivascular carotid electric injury (CI), which is particularly tailored to study aspects of endothelial re-endothelialization (Carmeliet et al, 1997).

# Results

## Endothelial proliferation coincides with CX$_3$CL1 induction

We first analyzed the endothelial monolayer in whole-mount carotids en face with confocal microscopy at different time points after CI. During the first 2 days after CI, the endothelial wound remained static and EC in the wound border appeared regularly shaped and densely aligned in an orderly fashion (Fig 1A–C). After day 2, a switch to a dynamic phase of endothelial wound closure occurred, which resulted in restoration of endothelial integrity at day 4 (Fig 1C). During the dynamic phase, EC appeared enlarged and irregular with filopodia extending into the wound (Fig 1A and B).

Endothelial cells proliferation is not observed during the first 36 h after injury but is noted after 50 h (Filipe et al, 2008). By Ki-67

staining, EC proliferation at day 2 was low and restricted to the immediate wound edge, but increased dramatically at day 3, due to an expansion of the proliferation area, and ceased with wound closure at day 4 (Fig 1D and E). Furthermore, proliferation was restricted to resident EC (Fig EV1C). This confirms earlier reports of EC proliferation during wound closure (Schwartz et al, 1978; Filipe et al, 2008), but also defines the temporal pattern of two distinct phases of endothelial regeneration, characterized by progressive endothelial proliferation.

Interestingly, during the static phase of EC regeneration, CX$_3$CL1 expression was induced but restricted to EC near the injury site (Fig 1F and H). However, CX$_3$CL1 expression area expanded markedly during the dynamic phase of endothelial regeneration, involving EC further away from the wound edge (Fig 1F and G), which recapitulated the spatiotemporal distribution of endothelial proliferation during EC regeneration.

## Recruitment of Ly6C$^{lo}$ monocytes during endothelial regeneration

CX$_3$CL1 expression after CI prompted us to investigate leukocyte recruitment in Cx3cr1$^{GFP/+}$ reporter mice, in which green fluorescent protein (GFP) is expressed in different intensities in Ly6C$^{hi}$ (GFP$^{lo}$) and Ly6C$^{lo}$ (GFP$^{hi}$) monocyte subsets (Jung et al, 2000; Auffray et al, 2007). In confocal microscopy, recruitment kinetics of GFP$^+$ cells to the EC wound mirrored that of CX$_3$CL1 expression and EC regeneration: a static phase of minimal recruitment on day 2, followed by a significant and progressive increase during the dynamic phase (Fig 2A and B). Furthermore, analysis of GFP intensities with different amplifier gain settings (Auffray et al, 2007) revealed that recruited GFP$^+$ cells were GFP$^{hi}$, suggesting recruitment of Ly6C$^{lo}$ monocytes (Figs 2C and EV1A). Additional in situ analysis revealed expression of CD11b but negative staining for NK1.1 or CD90.2, findings consistent with recruitment of Ly6C$^{lo}$ monocytes, but not NK or T cells (Fig EV1B). Furthermore, proliferation in the wound edge was restricted to EC and did not involve GFP$^+$ cells, suggesting progressive recruitment of GFP$^+$ cells during wound closure (Fig EV1C). Also, no GFP expression was observed in resident EC (Figs 2A and D, and 3A).

To investigate whether Ly6C$^{lo}$ monocytes are recruited to the endothelial monolayer, we scanned the entire dimension of the vessel wall along the z-axis by confocal microscopy and performed 3D reconstruction of the z-stack. This revealed that Ly6C$^{lo}$ monocytes are found only within the EC monolayer near the wound edge (Fig 2D).

---

**Figure 1.  Endothelial behavior in vivo after carotid injury.**

A   En face fluorescence microscopy of immunostained carotid arteries after CI. Scale bar: 400 μm.
B   Composite confocal images of endothelial wound edge. Scale bar: 75 μm.
C   Kinetics of wound closure post-CI (d0 n = 9, d1 n = 8, d2 n = 6, d3 n = 5, d4 n = 15).
D   Confocal images of immunostained carotids post-CI. Scale bar: 75 μm.
E   Kinetics of endothelial cell proliferation (d2 n = 24, d3 n = 12).
F   Composite confocal immunostained images after injury (left), scale bar: 150 μm. Single field images of individual channels (right), scale bar: 75 μm.
G   Quantitative analysis of CX$_3$CL1 expression in the injury area. n = 5 carotids/group.
H   Confocal images of healthy or injured endothelium, scale bar: 75 μm.

Data information: White arrows indicate the direction of flow of blood. Data are presented as mean ± SEM. Statistical analysis: (C, E) one-way ANOVA with Bonferroni's multiple-comparison test (***P < 0.001, *P < 0.05); (G) two-tailed Student's unpaired t-test, P < 0.0001.

**Figure 1.**

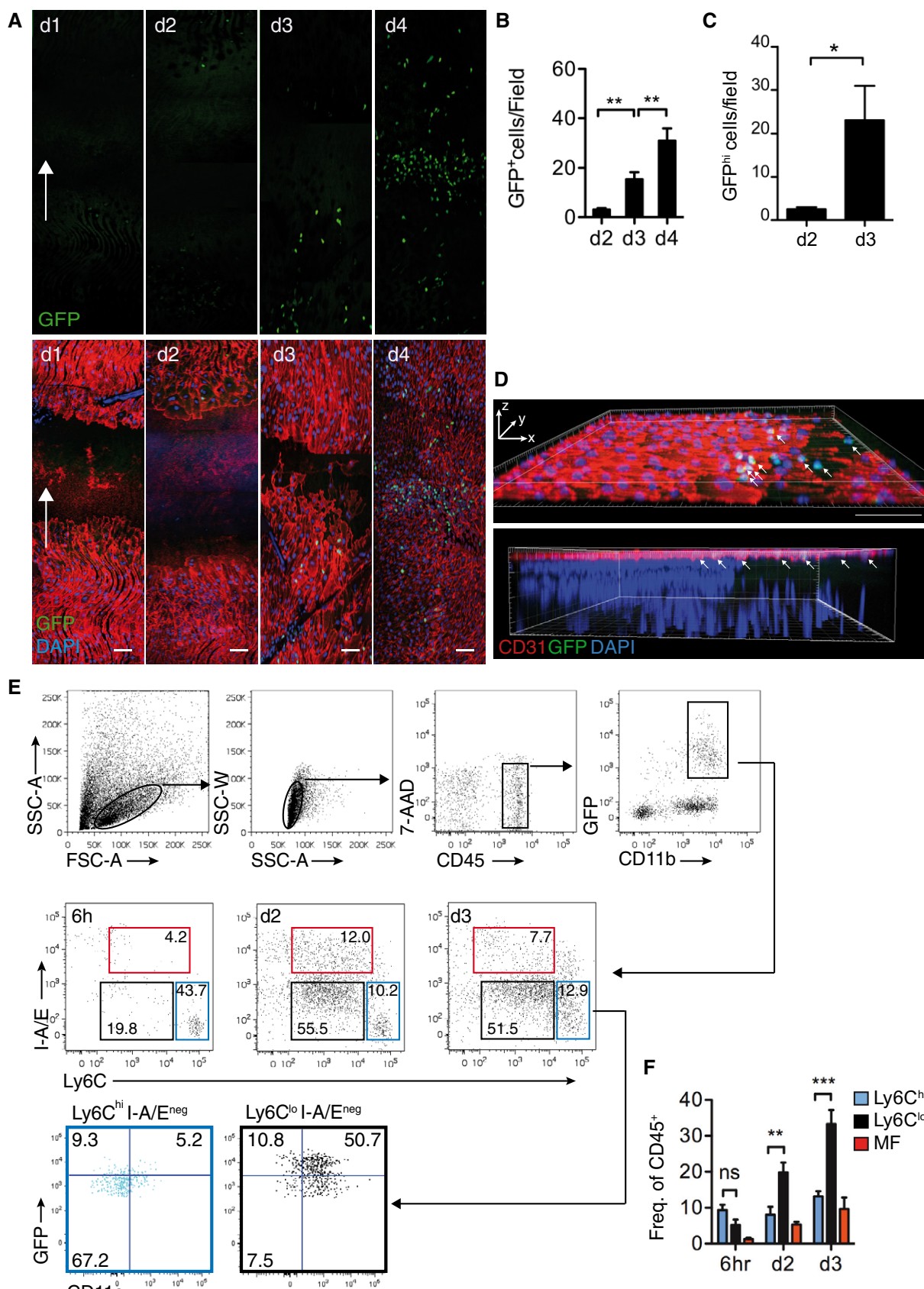

Figure 2.

**Figure 2. Ly6C$^{lo}$ monocyte recruitment after arterial injury.**

A   Composite confocal images of carotid arteries from *Cx3cr1$^{GFP/+}$* mice after injury. Top panel displays GFP channel. Scale bar: 75 μm.
B   Quantification of GFP$^+$ cells/field in the proximal wound (d2 $n = 15$, d3 $n = 12$, d4 $n = 10$).
C   Quantification of GFP$^{hi}$ cells/field after injury (d2 $n = 4$, d3 $n = 5$).
D   3D reconstruction of serial confocal images postinjury from *Cx3cr1$^{GFP/+}$* mice (d3). Scale bar: 75 μm. Arrows indicate dimensional axis.
E   Representative flow cytometry of single cell suspension from injured carotids (6 h $n = 3$, d3 $n = 5$, d3 $n = 3$).
F   Quantitative analysis of cell populations, depicted as frequency of CD45$^+$PI$^-$ (live, 6 h $n = 3$, d3 $n = 5$, d3 $n = 3$).

Data information: White arrows indicate the direction of flow of blood. Data are presented as mean ± SEM. Statistical analysis: (B, F) one-way ANOVA with Bonferroni's multiple-comparison test (***$P < 0.001$, **$P < 0.01$); (C) two-tailed Student's unpaired $t$-test, *$P = 0.0314$.

To more precisely characterize the phenotype and recruitment kinetics of leukocyte populations, we performed serial flow cytometric analysis from carotids of *Cx3cr1$^{GFP/+}$* mice (Fig 2E; Galkina et al, 2006; Gamrekelashvili et al, 2016). Neutrophils were recruited early after carotid injury (6 h) and declined rapidly thereafter (Fig EV2A and B). Recruitment of Ly6C$^{hi}$ monocytes and macrophages remained steady at low levels. In contrast, homing of Ly6C$^{lo}$ monocytes, defined as Ly6C$^{lo}$/MHC II$^{neg}$/CD11c$^+$/GFP$^{hi}$ (see Appendix Table S3), progressively increased in injured carotids over time (Figs 2E and F, and EV2C). Thus, our data demonstrate that the dynamic phase of endothelial regeneration is characterized by coordinate induction of CX$_3$CL1 and recruitment of Ly6C$^{lo}$ monocytes to the injured endothelium.

**Monocyte subsets regulate EC proliferation and regeneration**

To further investigate the relevance of CX$_3$CL1-mediated monocyte recruitment for EC regeneration *in vivo*, we studied mice with *Cx3cr1* loss of function (*Cx3cr1$^{GFP/GFP}$*), which have normal circulating monocyte levels but defective Ly6C$^{lo}$ monocytes recruitment to endothelial CX$_3$CL1 (Carlin et al, 2013). Following CI, recruitment of GFP$^+$ cells was strongly impaired in *Cx3cr1* mutant mice (Fig 3A and B). Impaired monocyte recruitment in *Cx3cr1*-deficient mice was accompanied by reduced EC proliferation (Fig 3C). Furthermore, the remaining EC wound area was markedly larger in *Cx3cr1*-deficient mice compared to controls, demonstrating defective EC regeneration after arterial injury in mice with *Cx3cr1* loss of function (Fig 3D). Importantly, these defects were rescued by adoptive transfer of sorted Ly6C$^{lo}$ monocytes from *Cx3cr1$^{GFP/+}$* donor mice, which increased the number of GFP$^+$ monocytes at the wound site (Fig 3E) and promoted endothelial wound closure after CI (Fig 3F).

To test whether monocyte subsets regulate EC proliferation, we isolated human classical (CD14$^{++}$CD16$^{neg}$) and non-classical (CD14$^+$CD16$^{++}$) monocytes from peripheral blood (Fig EV3A), which show conserved phenotypic and functional characteristics including CX$_3$CR1 expression (Fig EV3B; Ingersoll et al, 2010), and co-cultured them with human aortic EC (HAEC) pretreated with TNF-α/IFN-γ to induce CX$_3$CL1 (Fig EV3C; Schulz et al, 2007). While classical monocytes did not induce EC proliferation to a significant extend, non-classical monocytes approximately doubled the EC proliferation rate, which also occurred when direct cell contact was prevented by a Transwell insert (Fig 4A and B). In addition, EC proliferation was induced by addition of medium conditioned with supernatants from co-culture experiments (Fig 4C), which together suggested a secreted factor mediating EC proliferation. Indeed, monocytes cultured in the presence of EC showed increased levels of vascular endothelial growth factor A (VEGFA)

irrespective of EC-contact (Fig 4D). Importantly, addition of a VEGFA neutralizing antibody abrogated induction of EC proliferation by monocytes (Fig 4E). Furthermore, increased *Vegfa* expression was also confirmed in bona fide Ly6C$^{lo}$ monocytes compared to Ly6C$^{hi}$ monocytes (Fig 4F).

To test the capacity of human non-classical monocytes to promote EC regeneration, we performed a hybrid adoptive transfer experiment in which human monocytes are transferred into immunocompromised nude mice after CI. The effect on endothelial wound healing was measured by Evans blue injection, which stains the de-endothelialized region intensely and uniformly blue (Carmeliet et al, 1997; Sorrentino et al, 2007). Interestingly, also in the nude mouse model, the principle separation into a static and dynamic phase during endogenous EC regeneration was conserved, the dynamic phase starting after d3 postinjury (Fig 4G). Importantly, adoptive transfer of human non-classical monocytes improved endothelial wound closure during EC regeneration (Fig 4H). These results demonstrate that patrolling monocytes promote endothelial regeneration in the arterial circulation after injury.

# Discussion

Endothelial cells are quiescent in normal arteries but start to proliferate and replicate after vascular injury, which is required for re-endothelialization of larger endothelial wounds to prevent neointima formation or clotting (Schwartz et al, 1978; Haudenschild & Schwartz, 1979; Reidy & Schwartz, 1981).

We here show that the transition from a static phase to a dynamic phase of re-endothelialization is characterized by strong induction of endothelial CX$_3$CL1 expression and consecutive recruitment of *Cx3cr1*-expressing Ly6C$^{lo}$ monocytes into the endothelial monolayer. Notably, endothelial CX$_3$CL1 induction was also observed in a related vascular wire injury model (Liu et al, 2006). CX$_3$CL1 is a unique ligand for CX$_3$CR1 and acts as an adhesion molecule that promotes the firm adhesion of CX$_3$CR1-expressing monocyte subsets to EC (Fong et al, 1998; Ancuta et al, 2003). Indeed, the membrane-bound form of CX$_3$CL1 is predestined to serve as a homing cue under physiological flow, since it captures leukocytes without requiring selectin-mediated rolling or activation of integrins, while secreted chemokines are carried away with the blood stream (Bazan et al, 1997; Imai et al, 1997). Currently, the regulation of CX$_3$CL1 expression *in vivo* is unknown, but might involve inflammatory stimuli, such as TNF-α, IFN-γ, or TLR7 activation, which are involved in the vascular injury response and upregulate CX$_3$CL1 *in vitro* and *in vivo* (Zimmerman et al, 2002; Ahn et al, 2004; Schulz et al, 2007; Carlin et al, 2013).

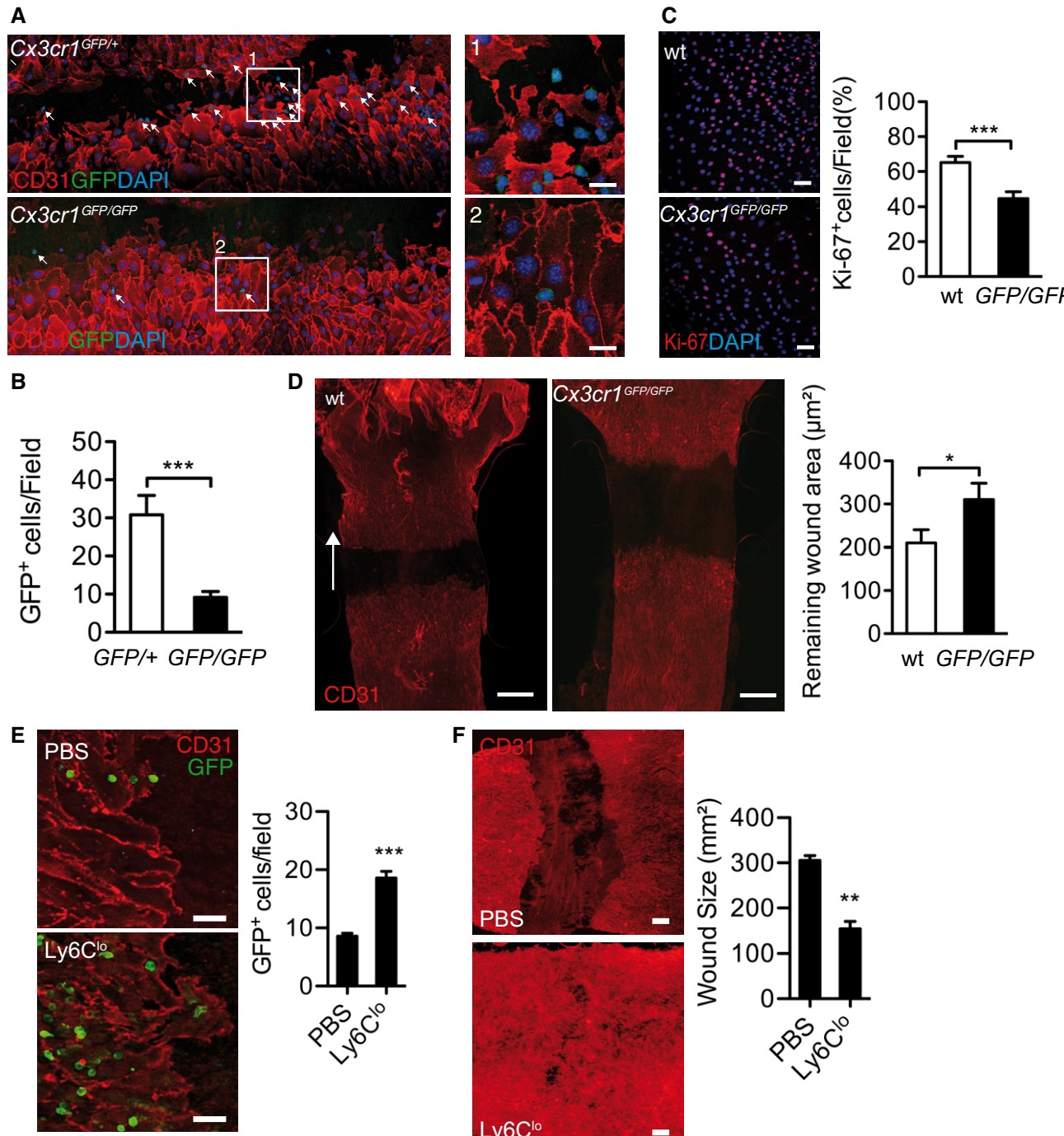

**Figure 3. *Cx3cr1* loss of function impairs EC regeneration.**

A   Composite confocal images of immunostained carotid arteries after injury (left), single close-up of insert (right). Scale bar: 75 μm. Arrows point to GFP⁺ cells.

B   Quantification of GFP⁺ cells/field in the proximal wound area (n = 10/12).

C   Representative immunostained images of proximal wound (left), scale bar: 75 μm. Quantification of EC proliferation (Ki-67⁺ cells/DAPI⁺ cells/field, right). n = 12/18.

D   EC wound healing in WT and *Cx3cr1*$^{GFP/GFP}$ mice (d3). Scale bar: 400 μm. White arrow indicates the direction of flow of blood. Representative immunostained image (left) and quantification of remaining wound area (between two wound fronts, right). WT n = 16, *Cx3cr1*$^{GFP/GFP}$ n = 9.

E   Representative confocal images of wound edge (left) and quantification of GFP⁺ cells (right) after injection of PBS or 1 × 10₆ Ly6C^lo monocytes from *Cx3cr1*$^{GFP/+}$ donors into *Cx3cr1*$^{GFP/GFP}$ recipients after CI. n = 4/4. Scale bar: 200 μm.

F   Representative immunostained images of wound area (left) and quantification of wound size (right) after injection of PBS or Ly6C^lo monocytes. n = 4/4. Scale bar: 200 μm.

Data information: Data are presented as mean ± SEM. Statistical analysis: (B–F) two-tailed Student's unpaired t-test. (B, C, E) ***P < 0.0001, (D) *P = 0.0473, (F) **P = 0.0015.

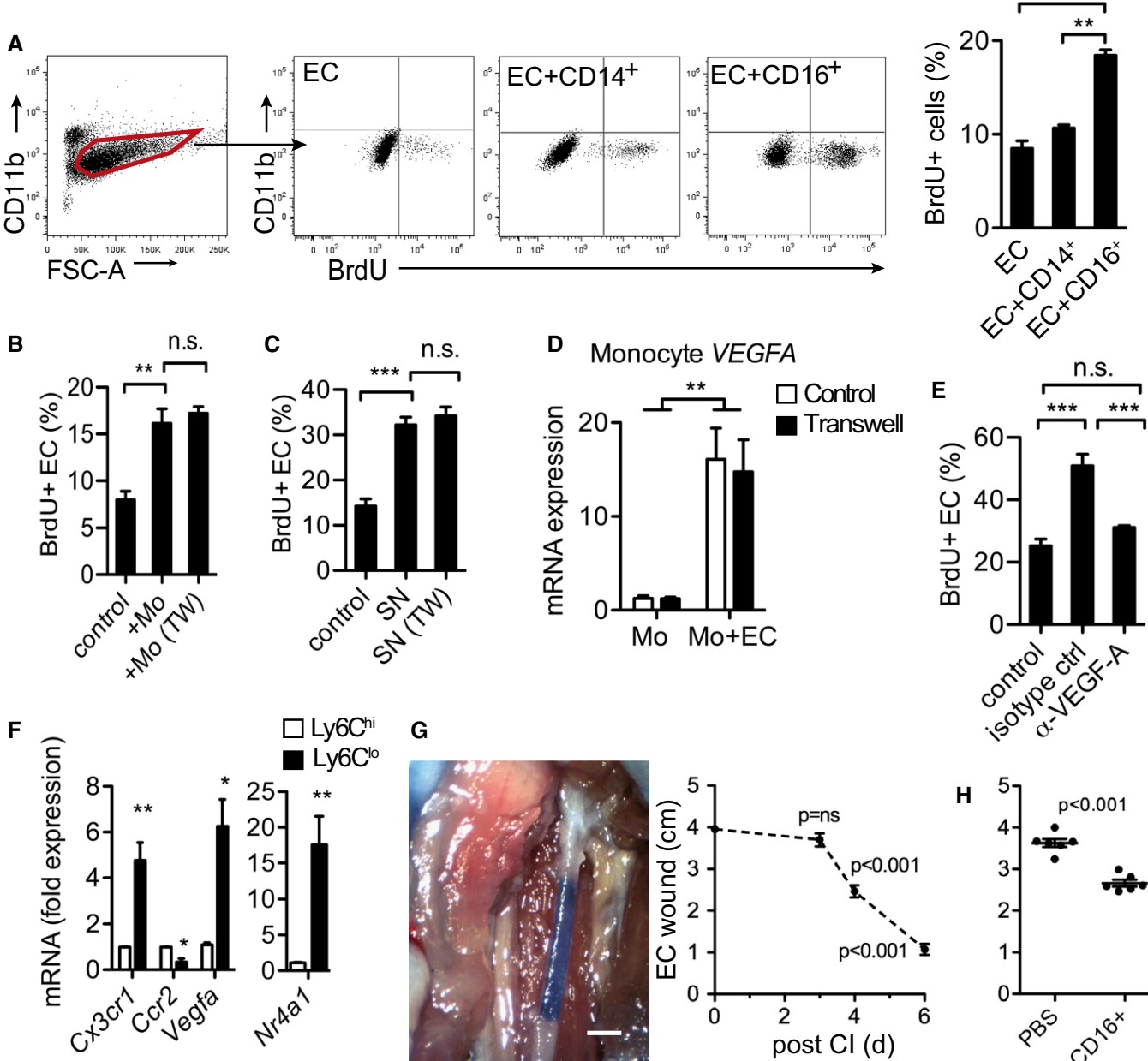

**Figure 4. Monocytes induce EC proliferation and regeneration.**

A    Representative flow cytometry analysis of EC BrdU incorporation after co-culture with human monocytes (left) and quantification of EC proliferation (right). *n* = 3.
B    EC proliferation after co-culture with non-classical monocytes with or without Transwell insert (TW), *n* = 3.
C    EC proliferation with conditioned medium (SN) from co-cultures generated with or without Transwell insert (TW), *n* = 3.
D    Gene expression by quantitative RT–PCR from non-classical monocytes cultured with or without EC for 24 h, with or without Transwell insert, normalized to input monocytes. *n* = 3.
E    EC proliferation after 3 days co-culture with non-classical monocytes with VEGFA neutralizing antibody or isotype control, *n* = 3.
F    Quantitative RT–PCR of murine monocyte subsets, normalized to gene expression of Ly6C^hi monocyte expression levels, *n* = 3.
G    Representative image (left) and quantification of EC wound closure (right) after CI in nude mice analyzed with Evans blue staining. Scale bar: 1 cm, *n* = 3 per group.
H    Quantification of wound closure after CI and transfer of non-classical monocytes (5 × 10^5 cells/mouse). *n* = 6/6.

Data information: Data are presented as mean ± SEM. Statistical analysis: (A–C, E, G) one-way ANOVA with Bonferroni's multiple-comparison test; (D) two-way ANOVA with Bonferroni's multiple-comparison test; ***$P$ < 0.001, **$P$ < 0.01, *$P$ < 0.05. (F, H) Two-tailed unpaired Student's *t*-test, *Cx3cr1*: **$P$ = 0.0014, *Ccr2*: *$P$ = 0.0254, *Vegfa*: *$P$ = 0.0238, *Nr4a1*: **$P$ = 0.0017.

Our unbiased analysis of myeloid cell subsets revealed that the immediate response to endothelial injury is dominated by neutrophils, which is in line with previous reports (Welt *et al*, 2000). The dynamic phase of EC regeneration, however, was characterized by recruitment of *Cx3cr1*-expressing Ly6C^lo monocytes, identified by flow cytometry and *in situ* confocal imaging based on prototypical

profiles (Ingersoll *et al*, 2010; Gamrekelashvili *et al*, 2016; Krishnasamy *et al*, 2017). Several lines of evidence suggest that Ly6C$^{lo}$ monocytes and their human counterparts, non-classical monocytes, are critically involved in arterial EC wound healing after injury. Recruitment of Ly6C$^{lo}$ monocytes steadily and significantly increased during the course of endothelial wound healing, coinciding with the onset of proliferation, while Ly6C$^{hi}$ monocyte recruitment remained constant, but on a much lower level. Both mouse and human monocyte subsets express high levels of CX$_3$CR1, which mediates firm attachment to CX$_3$CL1-expressing EC after injury (Carlin *et al*, 2013), and endothelial wound healing was impaired in *Cx3cr1*-deficient mice that have normal circulating levels of monocytes subsets (Auffray *et al*, 2009). Ly6C$^{lo}$ monocytes promote EC wound healing by recruiting neutrophils to mediate clearance of injured EC (Carlin *et al*, 2013) or by secreting growth factors like VEGF (Nahrendorf *et al*, 2007). Indeed, we show that mouse and human patrolling monocyte subsets express VEGF, which is required for induction of EC proliferation *in vitro*. Finally, human non-classical human monocytes improved EC wound closure in an adoptive transfer model, indicating that the regenerative potential of patrolling monocytes is conserved, which warrants further investigations as cell-based therapeutics to repair endothelial wounds in damaged arteries.

## Materials and Methods

Additional materials and methods are listed in the Appendix.

### Animals and surgical procedures

*Cx3cr1$^{GFP/+}$* reporter mice have been described previously (Jung *et al*, 2000). Mice were housed under specific pathogen-free conditions, and age- and sex-matched wild-type C57BL/6J mice or littermates were used as controls. Animal experiments were approved by the local animal welfare board. Carotid perivascular electric injury was performed with the following modification as previously described (Carmeliet *et al*, 1997; Sorrentino *et al*, 2007). Ten- to 15-week-old male mice were anaesthetized with a mixture of ketamine (80 mg/kg), xylazine (2.5 mg/kg), and midazolam (2.5 mg/kg) injected i.p., and the fur of the neck area was shaved completely. Bepanthen eye cream (Bayer) was applied to the eyes to prevent desiccation and unnecessary damage. In order to avoid hypothermia, the animals were placed on a heating pad and kept at a constant 37°C. An opening incision of 1–2 cm was placed along the midline of the neck, and a small portion of the left distal common carotid artery was laid free. Endothelial injury was induced with bipolar forceps (Cat. 20195-066, Erbe Elektromedizin GmbH) placed on the blood vessel. An injury was made through an electric impulse applied (intensity 2 W, bipolar mode) once for 2 s using Erbe microregulator (Erbotom ICC 50). The incision was sutured using polyester thread (polyester-S, 2xDRT 12, 6/0 USP, Catgut GmbH).

### Human endothelial cell culture and monocyte isolation

The experiments conform to the principles set out in the WMA Declaration of Helsinki and the Department of Health and Human Services Belmont Report and were approved by the local ethics board. Human CD14$^{++}$CD16$^{neg}$ and CD14$^{+}$CD16$^{++}$ monocytes were isolated from blood rings of healthy donors after written consent (Blood bank, Medizinische Hochschule Hannover) using CD14 microbeads and CD16 monocyte isolation kit (Miltenyi Biotec), respectively, according to manufacturer's instructions. The purity of the isolated cells was more than 90% routinely tested by flow cytometry. Human aortic endothelial cells (HAEC) were purchased from Lonza and cultured as per manufacturer's instructions.

### Statistical analysis

All data are presented as mean ± SEM. Significance of differences was calculated using unpaired, two-tailed Student's *t*-test. For comparison of multiple experimental groups, one-way ANOVA and Bonferroni's multiple-comparison test were used. *P*-values of less than 0.05 were considered to be significant and are indicated by the asterisk (***$P < 0.001$, **$P < 0.01$, *$P < 0.05$).

**Expanded View** for this article is available online.

### Acknowledgements

We thank Stefan Sablotny for excellent technical assistance, the Research Core Facilities Cell Sorting for support. This research has been funded by grants from Deutsche Forschungsgemeinschaft (Li948-5/1 and Ga2443/2-1) and BMBF (01GU1105B).

### Author contributions

TG, KK, AL, TK, CH, and LCN performed experiments. TG, KK, JG, AL, CH, and FPL analyzed the data. KK, TG, JG, JB, HH, and FPL wrote and edited the manuscript. FPL conceived the study.

### Conflict of interest

The authors declare that they have no conflict of interest.

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
