## [Review Process File · EMBO Molecular Medicine]

The chemokine receptor CX3CR1 coordinates monocyte recruitment and endothelial regeneration after arterial injury

Tobias Getzin, Kashyap Krishnasamy, Jaba Gamrekeshvili, Tamar Kapanadze, Anne Limbourg, Christine Häger, L. Christian Napp, Johann Bauersachs, Hermann Haller and Florian P. Limbourg

Review timeline:

Submission date:	04 January 2017
Editorial Decision:	15 March 2017
Revision received:	02 November 2017
Editorial Decision:	08 November 2017
Revision received:	15 November 2017
Accepted:	17 November 2017

Editor: Céline Carret

Transaction Report:

1st Editorial Decision

15 March 2017

I cannot apologise enough for the incredible delay that occurred with your article. I thank you for your patience and understanding. As I mentioned earlier, I have obtained one review and asked an advisor to offer another input on your article.

You will see from the set of comments pasted below that both are rather enthusiastic about the data but referee 2 would like to see more functional details and suggests experiments to do so. This referee also highlights *in vivo* experiments that could be performed to increase the clinical and translational relevance of the findings and we would like to encourage you to address these experimentally as much as possible. The advisor found the study convincing and of high quality. Despite a somehow limited novelty due to the paper by Becher et al. (In *J Cardiol* 2014), our advisor recommends publication pending revision.

We would therefore welcome the submission of a revised version for further consideration and depending on the nature of the revisions, this may be sent back to the referees for another round of review. Please note that it is EMBO Molecular Medicine policy to allow only a single round of revision and that, as acceptance or rejection of the manuscript will depend on another round of review, your responses should be as complete as possible.

Revised manuscripts should be submitted within three months of a request for revision; they will otherwise be treated as new submissions, except under exceptional circumstances in which a short extension is obtained from the editor.

I look forward to seeing a revised form of your manuscript as soon as possible. And again, I am truly sorry that it took so long to get back to you with this answer.

***** Reviewer's comments *****

Referee #2 (Remarks):

The authors explore the role of CX3CR1⁺ monocytes in endothelial regeneration after arterial injury using lineage tracing and loss of function strategies in mouse models.

They convincingly demonstrate that there are two phases of regeneration, and that endothelial cells at the wound edge and around, expressing the cytokine CX3CL1, undergo active proliferation and recruit monocytes expressing the receptor CX3CR1, in particular the Ly6Clo ones. They also show that in a mouse LOF for CX3CR1 there is impairment of monocyte recruitment and endothelial regeneration.

This is an interesting description of the phenomenon and shed some light on the CX3CL1-CX3CR1 in wound closure. Nevertheless there is a lack of detail on the molecular mechanism of this phenomenon.

The author went on to the human counterpart to investigate with subset of monocyte regulates EC proliferation in vitro.

Why this experiment has not been performed in the same species using the sorted CX3CR1 GFPlo and hi monocytes and murine EC? In this setting specific recruitment of monocytes in in vitro assay (boyden chamber or similar) could be performed, also using blocking Ab. And it would be interesting to assess which factors secreted by monocytes (or membrane bound) is responsible for inducing EC proliferation and wound closure. Selected factor could also be studied in vivo (expression and in some cases LOF-GOF strategies could be approached). This could have a more clinical relevance since specific signaling pathways could be targeted to improve/accelerate wound closure.

In the following set of experiments they transplanted human monocytes in immunocompromised mice, showing a more rapid wound closure. A rescue experiments in CX3CR1ko mice using wt subtypes of mouse monocytes would have been more informative.

1st Revision - authors' response

02 November 2017

Referee #2 (Remarks):

The authors explore the role of CX3CR1⁺ monocytes in endothelial regeneration after arterial injury using lineage tracing and loss of function strategies in mouse models. They convincingly demonstrate that there are two phases of regeneration, and that endothelial cells at the wound edge and around, expressing the cytokine CX3CL1, undergo active proliferation and recruit monocytes expressing the receptor CX3CR1, in particular the Ly6Clo ones. They also show that in a mouse LOF for CX3CR1 there is impairment of monocyte recruitment and endothelial regeneration.

This is an interesting description of the phenomenon and shed some light on the CX3CL1-CX3CR1 in wound closure. Nevertheless there is a lack of detail on the molecular mechanism of this phenomenon.

We completely agree with the reviewer that the molecular mechanisms are important. However, we first would like to give to consideration that the function of Ly6C^{lo} monocytes still is quite enigmatic, and that the principle demonstration of a specific effect on endothelial proliferation and wound closure is of interest to the field. Nevertheless, as described below, we have identified VEGF-A as one mechanism by which monocytes induce EC proliferation.

The author went on to the human counterpart to investigate with subset of monocyte regulates EC proliferation in vitro.

Why this experiment has not been performed in the same species using the sorted CX3CR1 GFPlo and hi monocytes and murine EC?

We chose the human co-culture system for several reasons:

First, human arterial EC, the focus of our current report, can be cultured reliably with high population homogeneity and stability, while we did not succeed to generate mouse EC cultures from aorta in sufficient quality and numbers using various protocols. Since the human and mouse monocytes subsets show conserved phenotypic and functional characteristics, we would propose that the human co-culture system is relevant for the context of our study.

A representative experiment showing flow cytometry data is shown for the reviewer below. We

compared EC cultures generated from microvascular EC (Lung) and from arterial EC (aorta). The initial EC purity after isolation (P0) was comparable between both cell culture types (EC content 68% vs 76%), although absolute yield of arterial EC was lower by a factor of 5, even when three mice were pooled for the isolation. However, after two passages to generate sufficient numbers for proliferation assays, the purity in the arterial EC culture dropped to 12%.

Furthermore, we show that human non-classical monocytes express high levels of VEGFA and induce EC proliferation in a VEGFA-dependent manner, and that high VEGFA expression is conserved in the equivalent mouse monocyte subset. Lastly, we would suggest that a human dataset on stimulation of EC proliferation by monocytes increases the relevance of the current study for the biomedical community.

In this setting specific recruitment of monocytes in in vitro assay (boyden chamber or similar) could be performed, also using blocking Ab. And it would be interesting to assess which factors secreted by monocytes (or membrane bound) is responsible for inducing EC proliferation and wound closure. Selected factor could also be studied in vivo (expression and in some cases LOF-GOF strategies could be approached). This could have a more clinical relevance since specific signaling pathways could be targeted to improve/accelerate wound closure.

As suggested we have performed a combination of transwell co-culture experiments, culture with conditioned medium, and blocking antibody studies. We thus identify that induction of EC proliferation by the patrolling monocyte subset is mediated by a secreted factor also present in supernatants from co-culture experiments. Monocytes express high levels of VEGFA when co-cultured with EC, and a VEGFA-blocking antibody neutralized the induction of EC proliferation by monocytes. Furthermore, mouse Ly6C^{lo} monocytes express significantly higher levels of Vegfa than Ly6C^{hi} monocytes. Together, these data demonstrate that patrolling monocytes induce EC proliferation via VEGFA. We show the data in revised Fig. 3C, D, E, F and describe the data in the abstract and results section.

In the following set of experiments they transplanted human monocytes in immunocompromised mice, showing a more rapid wound closure. A rescue experiments in CX3CR1ko mice using wt subtypes of mouse monocytes would have been more informative.

Following this critical suggestion we have performed a rescue study with Ly6C^{lo} monocytes sorted from *Cx3cr1*^{GFP/+} donor mice into *Cx3cr1*^{GFP/GFP} mutant mice after CI. This increased the number of GFP⁺ cells near the EC wound edge and enhanced EC wound closure compared to PBS injection. Thus, the *Cx3cr1* loss of function phenotype is rescued by acute transfer of normal monocytes, which supports the key finding of our report that CX3CR1 mediates monocyte recruitment and endothelial regeneration after arterial injury. This also suggests a therapeutic potential of monocyte transfer in the setting of larger-scale arterial injury. The new data set is shown in Fig. 3E, F and described in the abstract and results section.

2nd Editorial Decision

8 November 2017

Thank you for the submission of your revised manuscript to EMBO Molecular Medicine. We have now received the enclosed report from the referee who was asked to re-assess it. As you will see the reviewer is now supportive and I am pleased to inform you that we will be able to accept your manuscript pending the following final amendments:

***** Reviewer's comments *****

Referee #2 (Remarks for Author):

The revision of the work has adequately addressed my concerns

Corresponding Author Name: Francesco Saverio Tedesco and Giulio Cossu

Journal Submitted to: EMBO MOLECULAR MEDICINE

Manuscript Number: EMM-2016-07284-V2-Q